# A High-K^+^ Affinity Transporter (HKT) from *Actinidia valvata* Is Involved in Salt Tolerance in Kiwifruit

**DOI:** 10.3390/ijms242115737

**Published:** 2023-10-30

**Authors:** Shichao Gu, Shiming Han, Muhammad Abid, Danfeng Bai, Miaomiao Lin, Leiming Sun, Xiujuan Qi, Yunpeng Zhong, Jinbao Fang

**Affiliations:** National Key Laboratory for Germplasm Innovation & Utilization of Horticultural Crops, Zhengzhou Fruit Research Institute, Chinese Academy of Agricultural Sciences, Zhengzhou 450009, China; gushichao0906@163.com (S.G.); hanshiliang888@163.com (S.H.); muhammadabid@lsbg.cn (M.A.); baidanfeng1993@163.com (D.B.); linmiaomiao@caas.cn (M.L.); sleiming@163.com (L.S.); qixiujuangaoxing@163.com (X.Q.)

**Keywords:** ion transport, HKT, overexpression, salt tolerance, *A. valvata*

## Abstract

Ion transport is crucial for salt tolerance in plants. Under salt stress, the high-affinity K^+^ transporter (HKT) family is mainly responsible for the long-distance transport of salt ions which help to reduce the deleterious effects of high concentrations of ions accumulated within plants. Kiwifruit is well known for its susceptibility to salt stress. Therefore, a current study was designed to decipher the molecular regulatory role of kiwifruit HKT members in the face of salt stress. The transcriptome data from *Actinidia valvata* revealed that salt stress significantly induced the expression of *AvHKT1*. A multiple sequence alignment analysis indicated that the AvHKT1 protein contains three conserved amino acid sites for the HKT family. According to subcellular localization analysis, the protein was primarily present in the cell membrane and nucleus. Additionally, we tested the *AvHKT1* overexpression in ‘Hongyang’ kiwifruit, and the results showed that the transgenic lines exhibited less leaf damage and improved plant growth compared to the control plants. The transgenic lines displayed significantly higher SPAD and *Fv*/*Fm* values than the control plants. The MDA contents of transgenic lines were also lower than that of the control plants. Furthermore, the transgenic lines accumulated lower Na^+^ and K^+^ contents, proving this protein involvement in the transport of Na^+^ and K^+^ and classification as a type II HKT transporter. Further research showed that the peroxidase (POD) activity in the transgenic lines was significantly higher, indicating that the salt-induced overexpression of *AvHKT1* also scavenged POD. The promoter of *AvHKT1* contained phytohormone and abiotic stress-responsive *cis*-elements. In a nutshell, AvHKT1 improved kiwifruit tolerance to salinity by facilitating ion transport under salt stress conditions.

## 1. Introduction

Kiwifruit is an economic fruit from the genus *Actinidia* that contains 54 species, 52 of which originated in China. Generally, kiwifruit-producing areas in northern China have salinized soil, which greatly affects plant growth, fruit yield, and quality. Presently, China’s kiwifruit industry lacks salt-tolerant rootstocks, which restricts the healthy development of the industry. Salt stress inhibits plant growth primarily by increasing Na^+^ contents and causing ion toxicity. The salt-tolerant plant cells mainly improve salt tolerance by reducing the amount of Na^+^ in their cytoplasm and increasing the flow of toxic ions throughout their tissues. Thus, maintaining a high K^+^/Na^+^ ratio is important for salt tolerance.

Plants are more likely to be salt tolerant when inorganic ions are transported between tissues. In plants, High-affinity K^+^ transporter (HKT) plays a significant role in maintaining K^+^ and Na^+^ homeostasis, which is well documented. Maintaining high K^+^/Na^+^ levels in stems and leaves of glycophyte can alleviate the harm caused by salt stress [1]. Na^+^ is transported between plant tissues via the HKT protein found in root and xylem epidermal cells. There are two types of HKT proteins: type I, which specifically transports Na^+^, and type II, which transports Na^+^ and K^+^ [2,3]. The conserved amino acid sequences of HKTs may play a significant role in determining their functional differences in plants. The structural analysis of *AtHKT1;1* predicted the existence of four transmembrane channels containing unique serine and glycine sites that ensure functional conservation [4]. HKT proteins are classified as SGGG-type and GGGG-type based on functional distinctions. Both the wheat and *Arabidopsis* [5] demonstrated that the conserved glycine motif in the transmembrane structure is critical to the protein’s capacity to transport K^+^ [6,7]. The rice *OsHKT2;1* is an SGGG-type transporter with Na^+^ and K^+^ transport activity [8,9]. Furthermore, external Na^+^ concentration affects how wheat ion transporters absorb Na^+^ and K^+^ ions [10]. Therefore, the conserved glycine sites in the HKT transporter will influence the type of transported ions, but it does not inherently determine the transport results of plants under stress conditions.

The HKT gene has been reported to improve plant salt tolerance by maintaining ion homeostasis. The transcriptional modifications of Salt overly sensitive (*SOS)*, *HKT,* and Na^+^/H^+^ antiporters (*NHX)* family members caused salt-tolerant barley leaves to accumulate more K^+^, while salt-sensitive rice leaves accumulated more Na^+^ [11]. The high expression of *HvHKT1;5* in barley resulted in an increase in K^+^ concentration and a decrease in Na^+^/K^+^ [12]. The *OsHKT1;4* participated in the transport of Na^+^, which caused a decrease in Na^+^ content in rice shoots [13]. *AtHKT1;1* in *Arabidopsis* improves the salt tolerance of plants by transferring Na^+^ from roots to stems [14,15]. In the Na^+^ detoxification mechanism, AtHKT1;1 transports Na^+^ accumulated in the xylem vessels of the root system to the xylem parenchyma cells to reduce the amount of Na^+^ transported to the shoot, thereby protecting the leaves from salt stress [16]. Under salt stress, mutation and overexpression of *AtHKT1;1* in *Arabidopsis thaliana* reduced the Na^+^ and sugar metabolites in the stem [17], indicating that ion transport genes indirectly affected the process of metabolic pathways. The function of *AtHKT1* in *Arabidopsis* has been thoroughly studied. Our preliminary results suggested that the function of the HKT gene family in transporting ions may vary in plants depending on external conditions, i.e., external ion concentration and the salt stress regulation mechanism of plants. The introduction of *TsHKT1;2* into *Arabidopsis thaliana* regulated the transport of K^+^ under salt stress, thus improving the salt tolerance of plants [18]. Overexpression of *AtHKT1* in potatoes reduced the Na^+^ content in leaves, increased K^+^/Na^+^, and eventually reduced salt damage [19]. Gene mutation of wheat *TaHKT1;5-D*, increased the Na^+^ content in leaves of the transgenic lines, proving its role in regulating the transport of Na^+^ from plant roots to leaves [20]. The *ZmHKT2* improved the salt tolerance of maize by regulating the K^+^ content in stems [21]. The *HKT* gene was also found to be involved in ion transport by forward genetics. Through QTL mapping, it was found that *ZmHKT1* promoted the transport of Na^+^ from root to stem by increasing the Na^+^ content in xylem sap and reducing the Na^+^ content in leaves [22]. The *VisHKT1* mapped by major QTL in grapes mediated the transport of Na^+^ from scion to rootstock [23]. Also, the *VviHKT1;6–8* gene mapped by QTL in grapes was involved in the transport of Na^+^ in cells [24].

The HKT protein regulated the ion transport between different tissues. Overexpression of *GmHKT1* in soybeans enhanced the K^+^/Na^+^ in the roots and stems of transgenic plants, thus improving the salt resistance [25]. SvHKT1;1, a Na^+^-specific transporter from salt-tolerant Halophytic turf grass, was involved in the transport of Na^+^ from roots to shoots under salt stress to enhance the salt tolerance of plants [26]. CmHKT1;1 restricted the transfer of Na^+^ from roots to shoots in grafted cucumber seedlings and played a role in salt tolerance [27]. The HKT and NHX ion transporters were responsible for Na^+^ and K^+^ accumulation in sorghum under salt stress [28]. In addition, SbHKT1;4 from sorghum can transport Na^+^ and K^+^ under high Na^+^ conditions to maintain Na^+^ and K^+^ homeostasis under salt stress [29]. The HKT functioned differently in different species, such as CmHKT1;1 and VviHKT1;1, and improved salt tolerance by limiting the transport of ions from the root to the shoot, thereby reducing the effect of ions on the shoot tissue. However, *ZmHKT1*, *TaHKT1;5-D*, and *SvHKT1;1* improve salt tolerance by increasing the transport of Na^+^ in roots to shoots. Thus, we speculate that HKT may possess different ion transport functions in different plant species.

Promoter analysis can facilitate untangling the transcriptional regulation mechanism of a gene. The promoter of *EsHKT1;2* was transferred into *Arabidopsis AtHKT1;1* promoter mutant lines, which enhanced the salt tolerance of transgenic plants [30]. A comparison study of *TaHKT1;5* and *TmHKT1;5* showed that wild-type wheat possessed more binding elements (including Jasmonic acid and ABA binding elements) than cultivated wheat [31]. This also indicates that the hormone pathway may be implicated in the salt resistance process via binding to the promoter of the ion transport gene. A 1508 bp promoter sequence upstream of the *TmHKT1;4* in wheat responded to salt stress, dehydration stress, and ABA treatment [32]. It indicated that the HKT gene was affected by several other elements under abiotic stresses. OsMYBc in rice binds to the promoter of *OsHKT1;1* through the AAANATNC(C/T) site to reduce the Na^+^ content in phloem sap [33]. OsSIRH2-14 can interact with OsHKT2 to reduce the Na^+^ content in roots and improve salt tolerance in rice under a high salt environment [34]. The molecular regulatory network of HKT protein can be further revealed by studying the abiotic stress-specific binding sites in the upstream promoter region.

Previously, we reported that one genotype of *A. valvata* performed well under salt stress and had strong salt tolerance [35]. Possibly, HKT ion transporters positively regulate salt tolerance by controlling ion homeostasis. The function of HKT is speculated to be influenced by the following factors: (i) external ion concentrations; (ii) the specific structure of HKT proteins, based on conserved amino acid sites in the transmembrane region; and (iii) the differential mechanism of HKT between salt-tolerant and salt-sensitive plants. The molecular regulatory mechanism of salt tolerance in *A. valvata* is still largely unclear and demands further exploration. In the current study, we only screened one HKT protein named AvHKT1 in the transcriptome data of *A. valvata* under salt stress. A phylogenetic analysis showed the AvHKT1 has unique HKT amino acid sites. We reported that AvHKT1 positively regulates kiwifruit salt tolerance by regulating the transport of ions and accumulating peroxidase (POD) under salt conditions. Also, we analyzed the possible binding sites for Cis-element in the upstream promoter region of *AvHKT1*. The study reveals the regulatory function of *AvHKT1* for ion transport under salt stress. Additionally, the current study provides valuable insights into Na^+^ and K^+^ transport by AvHKT1 and deepens our understanding of the salt tolerance mechanism in kiwifruit.

## 2. Results

### 2.1. Phylogenetic Tree Analysis for AvHKT1

We previously investigated transcriptional and metabolic changes in *A. valvata* under salt stress which led to the identification of *AvHKT1*, an HKT family member. The 930 bp long genomic sequence of *AvHKT1* encoded 310 amino acids. Phylogenetic analysis showed that AvHKT1 had 57% to 100% amino acids identified in 16 HKT family members from different plant species (Figure 1). The AvHKT1 shared higher sequence similarity with AtHKT1;1 and TsHKT1;2. Additionally, AvHKT1 was closely related to CsHKT6 and PpHKT1 in citrus and peach, respectively.

### 2.2. Conserved Sites and Trans-Membrane Domains for AvHKT1

The conserved regions in a protein sequence determine its function. The conserved amino acid sites of four selectivity pore-forming regions (P-loops) may be related to the classification of HKT family proteins in *Arabidopsis* and other plant species. According to the sequence alignment results, the AvHKT1 had three of the four conserved amino acid sites (Figure 2), similar to the results reported for CsHKT6 in citrus. A transmembrane structure analysis showed that AvHKT1 has five transmembrane domains (Figure 3).

### 2.3. The Expression Pattern of AvHKT1

We measured the expression levels of *AvHKT1* in *A. valvata* under salt stress to understand how *AvHKT1* regulates salt tolerance. The expression levels of *AvHKT1* were significantly up-regulated in 72 h compared with no treatment (control). Additionally, *AvHKT1* was mainly expressed in roots, stems, and petioles (Figure 4b). The subcellular localization results showed that the empty vector (35S-GFP) had obvious fluorescence expression in the nucleus and cell membrane. The cell containing fusion plasmid also showed GFP fluorescence signals in the nucleus and cell membrane (Figure 5). Therefore, it was speculated that the AvHKT1 protein was localized in the cell membrane and nucleus.

### 2.4. Upstream Promoter Region Analysis of AvHKT1

The homologous cloning method was employed to amplify the 2550 bp upstream promoter sequence of *AvHKT1*. The cis-regulatory element results predicted several stress-responsive elements including ABRE (involved in abscisic acid reaction), ARE (involved in anaerobic reaction), and Box 4 (involved in light response), and hormone response elements including P-box (responsive to gibberellin), TGA-element (involved in auxin response), and CGTCA-motif and TGACG-motif (involved in MEJA-response signals) (Table 1). Therefore, we preliminarily speculated that the *AvHKT1* gene may be involved in hormone signaling and abiotic stresses.

### 2.5. Overexpression of AvHKT1 Enhances Salt Stress Tolerance of A. chinensis

Sequence alignment showed that AvHKT1 is related to AtHKT1 in *Arabidopsis thaliana* and TsHKT1 in *Thellungiella halophila*. A conserved structural amino acid site analysis showed that AvHKT1 has three conserved amino acid sites of the HKT family. The exon splicing affects the plant’s response to K^+^ and Na^+^ concentrations [36]. The promoter region of *AvHKT1* contained multiple abiotic stress response elements. Therefore, only the functional identification of transgenic lines can determine the specific family classification and function of AvHKT1 protein in kiwifruit.

We selected three positive overexpression lines for *AvHKT1* to investigate its function in kiwifruit under salt tolerance. The overexpression lines named OE1–3 were detected by PCR and the results showed that three lines had a distinct bright band for the target gene that the wild-type lines lacked (Figure 6a). Among the three positive lines, we identified 30, 28, and 29 plants, respectively. Later, plants with an identical growth status from each overexpression line were selected for functional verification.

The positive plants and wild-type seedlings were treated in an MS medium containing 1.0% NaCl. The samples were observed and sampled at 0 d, 1 d, and 7 d after treatment. After 7 days of salt stress, the functional validation assay for *AvHKT1* in kiwifruit overexpression lines revealed healthy growth, while WT plants suffered substantial damage (Figure 7a). The MDA content was significantly higher in wild-type plants compared to the overexpression lines, while POD activity was higher in overexpression lines compared to the WT plants (Figure 7b,c). Additionally, the SPAD value of the whole WT plant was lower than that of the overexpression lines (Figure 7d). The RT-qPCR analysis was performed to monitor the expression levels of *AvHKT1* in OE lines. The results showed that the expression levels of *AvHKT1* were significantly higher in the OE lines than in the WT plants (Figure 7e).

The HKT protein has been reported to be involved in ion transport. The ion detection assay showed that OE lines had a lower Na^+^ concentration in the leaves than the WT after 7 days of salt stress (Figure 8a). However, no differences were detected for the K^+^ content in leaves between the wild-type and OE lines (Figure 8b). Additionally, the results showed that OE lines had lower Na^+^ concentrations and higher K^+^ concentrations than the wild-types (Figure 8c,d). The OE lines had a lower Na^+^/K^+^ ratio in both leaves and stem than the WT plants, indicating that *AvHKT1* improved the OE line’s performance under salt stress by regulating the homeostasis of Na^+^ and K^+^ (Figure 8e,f).

## 3. Discussion

Maintaining ion homeostasis is critical for the survival of plants during salt stress. HKT is involved in short-distance ion transport between cells as well as long-distance ion transport within plants [37]. There are only a few reports available on the function of this protein in kiwifruit. In the current study, kiwifruit HKT protein was significantly induced by salt stress. Kiwifruit HKT family members have conserved motifs comparable to those seen in other plants. A function verification assay showed that AvHKT1 improved salt tolerance by regulating the transport of Na^+^ and K^+^ in transgenic lines. The proteins that can possibly regulate the promoter region of *AvHKT1* include hormone-binding sites and stress response elements. Future research should examine the joint impact of the HKT protein and hormone signaling on the stress response pathway.

A classification analysis of family members can help researchers to predict their function. The phylogenetic analysis can clearly classify HKT family members into two distinct classes. However, identifying the classes for HKT family members cannot be used to establish how a gene will function. The TsHKT1;2 in *Thellungiella salsuginea* is a typical K^+^ affinity protein that improves salt tolerance by transporting K^+^ within the plant [18]. A phylogenetic analysis showed that TsHKT1;2 clustered more closely to *Arabidopsis* AtHKT1 (Figure 1). In addition, AtHKT1 and TsHKT1;2 were not clearly distinguished, despite the identification of two distinct classes for HKT family members. AvHKT1 was more closely related to HKT in citrus and peach and it grouped in the same class with AtHKT1 and TsHKT1;2. Therefore, the phylogenetic analysis may not be possible to predict the function of AvHKT1. We have previously described that class I HKT family proteins have classical S-G-G-G conserved amino acid sites in the P-loop region, and class II HKT protein families have G-G-G-G conserved sites. However, relying on this classification alone does not completely determine the protein’s transport ability. For instance, OsHKT2;1 has S-G-G-G conserved sites but is a type II transporter protein. The multiple sequence alignments of 17 HKTs from different plant species showed that there were only three conserved amino acid sites in the AvHKT1 protein sequence. The lack of the fourth conserved site in AvKHT1 may be caused by genomic variations between various species as a result of genome evolution processes. It has also been reported that exon splicing will influence ion homeostasis [36]. This might serve as a reference for our research on the role of AvHKT1 in kiwifruit salt tolerance. The same deletion also occurred in citrus, which speculated that the P4 conserved amino acid site may not be required as a gene function marker in kiwifruit.

The expression of an *HKT* family member under salt stress provides clear hints about their function. The expression of *AvHKT1* in *A. valvata* roots initially decreased and then increased under salt stress. This phenomenon was more likely caused due to the complex regulatory mechanism of the HKT protein under higher Na^+^ concentrations. The soil microorganism *B. subtilis* GB03 enhances Na^+^ tolerance in *Arabidopsis* by inducing the expression of *AtHKT1;1* above ground and suppressing *AtHKT1;1* in roots [38]. Similarly, studies of rice showed that *OsHKT1;1* was significantly up-regulated in stems after salt stress with little difference in the expression in roots [33]. Under salt stress, *PtHKT1* in salt-tolerant crops was significantly up-regulated in roots [39]. *A. valvata* showed enhanced tolerance under salt stress and the expression of *AvHKT1* in its roots was significantly higher than that in untreated roots (Figure 4b). Therefore, we concluded that *AvHKT1* can regulate ion transport in different tissues of plants. Additionally, we have discovered that AvHKT1 was a cell membrane-localized protein, suggesting that it may promote ion transport between cells, thereby conferring salt tolerance in plants. The OE lines were less affected compared to WT plants after 7 days of salt stress. The higher POD activity in OE lines under salt stress indicated that the overexpression of *AvHKT1* participated in the removal of peroxides in the plant under stress. The same mechanism has been found for oxygen scavenging by the overexpression of *PeHKT1;1* in Poplar [40]. The varying ion concentrations in stems and leaves indicated that the toxic ions were possibly transported from stem to leaves to regulate ion transport within the plant and thus mitigate stress injury under salt stress. The overexpression of *AvHKT1* greatly influences the ion content between different tissues within a plant.

The upstream promoter region of *AvHKT1* contained multiple hormone-binding elements, including auxin, gibberellin, and abscisic acid binding elements, suggesting that *HKT* genes may be involved in multiple pathways in response to abiotic stresses. Plant hormones act as signal transductors during plant growth and developmental processes as well as under stress conditions. The role of plant hormones under abiotic stresses is crucial. ABA and Jasmonic acid binding elements were found in the promoter region of *TaHKT1;5* in wheat [31]. In addition, the ABA signaling binding element was also found in the promoter region of *Triticum monococcum TmHKT1;4*, while the activity of this promoter region was activated by exogenous ABA treatment [41]. The role of ABA in plants under salt stress has been extensively reported in the past. Exogenous ABA treatment increased the activity of the *HKT* promoter inside the plant, indicating the perception of ABA signaling by the HKT promoter and demonstrating the association of the hormone pathway with the ion transport pathway under salt stress. In strawberries, salt stress treatment affected the ABA hormone metabolic pathway [42]. In wheat, ABA signaling affected the activity of *HKT* promoter elements. The *AvHKT1* promoter region in kiwifruit has an ABA binding site which can be explored for its regulation mechanism of salt tolerance in future research. The gibberellin signaling pathway can improve plant stress resistance by interacting with ABA and ethylene [43]. Abiotic stress can also affect the Jasmonic acid signaling pathway. In grapes, exogenous Jasmonic acid application was found to promote the growth of salt-sensitive plants and improve their resistance to salt stress [44]. The overexpression of *MdJAZ2* in *Arabidopsis* improved its tolerance to salt stress [45]. The AUX/IAA is also involved in the plant abiotic response. *VvIAA18* improved the salt tolerance of plants by increasing the SOD activity and reducing the H_2_O_2_ and MDA levels [46]. Plant hormones can improve plant resistance to salt stress, however, there are fewer reports on the association of hormone-regulatory pathways with ion transport pathways.

In addition, we identified binding sites for MYB, ABRE, G-box, abiotic stress, and light response signals in the promoter region of *AvHKT1*. The MYB and ABRE binding sites were also found in the upstream 760 bp promoter region of *AlHKT2;1* in *Aeluropus lagopoides* [47]. The regulatory role of MYB transcription factors for *HKT* genes has been verified in rice. The role of light responsiveness signaling in plant abiotic stress processes has also been reported. The G-box binding factor is a signal for blue light response in wheat, which regulates abiotic stress processes in plants by participating in the ABA signaling pathway [48]. Therefore, the prediction of the *AvHKT1* promoter element in this experiment can be used to uncover a deeper regulatory mechanism of this gene.

## 4. Materials and Methods

### 4.1. Plant Material and Stress Treatment

ZMH (a salt-tolerance genotype of *A. valvata*) from tissue culture was grown in a greenhouse at Zhengzhou Fruit Research Institute, Chinese Academy of Agriculture Science, Henan Province, China (Latitude 34°43′ N, Longitude 113°39′ E, and Altitude 111 m). To evaluate the salt tolerance, the ZMH plants at the 4–6 leaf stage were treated with 0.4% NaCl per net weight of the growing medium in the pot. Whole plants were collected at 0 h, 12 h, 24 h, and 72 h after salt treatment for subsequent analysis.

The transgenic and wild-type plants were separately grown on a Murashige and Skoog (MS) medium containing 0.72% agar, 3% sucrose, and 1.0% NaCl for function verification. The samples were taken at 0 h, 1 d, and 7 d after treatment. All samples were immediately frozen in liquid nitrogen and stored at −80 °C for further research.

### 4.2. Sequence Identification and Structure Analysis

We obtained the full-length sequences of kiwifruit AvHKT1 from the previously published data (PRJNA726156). The sequence of AvHKT1 was subjected to the InterProscan (http://www.ebi.ac.uk/Tools/pfa/iprscan/, accessed on 10 March 2023) and NCBI Conserved Domains (https://www.ncbi.nlm.nih.gov/cdd/?term=, accessed on 10 March 2023) databases to confirm the presence of conserved domains (PF02386). The protein sequences of *Arabidopsis thaliana*, *Citrus sinensis*, *Nicotiana tabacum*, *Oryza sativa* subsp. *Japonica*, *Prunus persica*, *Triticum aestivum*, *Solanum lycopersicum*, and *Thellungiella salsuginea* were retrieved from the Uniport database (accession numbers are shown in Table 2). The multiple sequence alignment analysis of all HKT sequences was carried out using the ClustalW method with default settings. The phylogenetic tree was built by using MEGA-X (Mega Limited, Auckland, New Zealand) with the neighbor-joining (NJ) method and 1000 bootstrap replicates [49]. The amino acid sequence alignment was shown by DNAMAN (version 9, Lynnon Biosoft, QC, Canada). The transmembrane structure of AvHKT1 was predicted using the Protter online prediction tool (http://wlab.ethz.ch/protter/start/, accessed on 15 March 2023).

### 4.3. Total RNA Extraction and RT-qPCR Analysis

Total RNA extraction for all samples was performed by using the Plant Total RNA purification kit (Waryoung, Beijing, China) following the manufacturer’s instructions. Genomic DNA was removed by DNase I (Thermos Scientific, Waltham, MA, USA) from the total RNA. First-strand cDNA was synthesized by using a commercial kit (TOYOBO, Tokyo, Japan). RT-qPCR primers were designed by using the Primer Premier 5.0 tool [50]. All the primers used in this study are listed in Appendix A. RT-qPCR analysis was performed as follows: 95 °C for 5 min; followed by 40 cycles of 95 °C for 10 s, 60 °C for 10 s, 72 °C for 10 s; and final extension at 65 °C for 1 min. Relative expression levels were calculated by using the 2^−∆∆Ct^ method [51].

### 4.4. Cloning CDS and Promoter Sequences of AvHKT1

The open reading frames (ORFs) of *AvHKT1* was cloned from the cDNA of the *A. valvata* root using a specific pair of primers. Similarly, The *AvHKT1* promoter sequence was also cloned from *A. valvata* genomic DNA by using a specific pair of primers. Cis-acting regulatory elements in the promoter regions of *AvHKT1* were identified by the PlantCARE database (https://bioinformatics.psb.ugent.be/webtools/plantcare/html/, accessed on 11 May 2023).

### 4.5. Subcellular Localization Analysis of AvHKT1

To determine the subcellular localization of AvHKT1, the full-length coding sequence of *AvHKT1* lacking the stop codon was amplified and cloned into the pAN580-GFP vector (4712 bp). Protoplasts for performing subcellular localization were generated from 4-weeks-old rice leaves. The isolation and transformation of protoplasts were performed by following a previously reported method [52]. The GFP signals in transformed leaves were detected under a laser scanning confocal microscope (Nikon C2-ER, Tokyo, Japan).

### 4.6. Transformation of Kiwifruit

The CDS of *AvHKT1* without a stop codon was fused to the PBI121 vector driven by the CaMV35S promoter. The recombinant plasmids were transformed into leaf discs of *A. chinensis* (a salt-sensitive cultivar ‘Hongyang’) according to a previously described method [53]. The transgenic plants were grown on an MS medium and positive lines were detected using RT-PCR methods. All the pairs of primers used in these experiments are listed in Appendix A.

### 4.7. Physiological Analysis

The chlorophyll contents were measured using a SPAD meter (SPAD 502 plus, Minolta, Japan) [54]. The Na^+^ and K^+^ ions and MDA content were detected according to the protocol of a commercial kit (Nanjing Jiancheng Bioengineering Institute, Nanjing, China). The activity of POD was also quantified by using a relevant kit (Solarbio, Beijing, China).

### 4.8. Statistical Analysis

All the experiments in this study were performed in three replicates. All the statistical analyses were performed by using SPSS 21.0 software (IBM Corporation, Chicago, IL, USA). The significance level of data was analyzed using one-way ANOVA and *t*-test. The mean differences were compared by Tukey’s multiple comparisons test at *p* < 0.05 or *p* < 0.01. All the results were presented as the means ± SE (standard error) and different letters indicated significant differences between means.

## 5. Conclusions

In summary, our results suggest that AvHKT1 is a cell membrane-localized protein with Na^+^ and K^+^ transport ability. In *A. valvata*, *AvHKT1* is mainly expressed in root, stem, and petiole. Overexpressed *AvHKT1* enhanced the ion transport and POD accumulation. Furthermore, the cis-regulatory elements prediction in the upstream promoter region of the *AvHKT1* gene speculated the relationship between the phytohormone transduction pathway and ion transport. Our result enhanced the understanding of the role of *AvHKT1* in kiwifruit salt tolerance. Future research in the salt stress context should be oriented around the association of the ion transport pathway and the plant hormone signaling pathway.

## Figures and Tables

**Figure 1 ijms-24-15737-f001:**
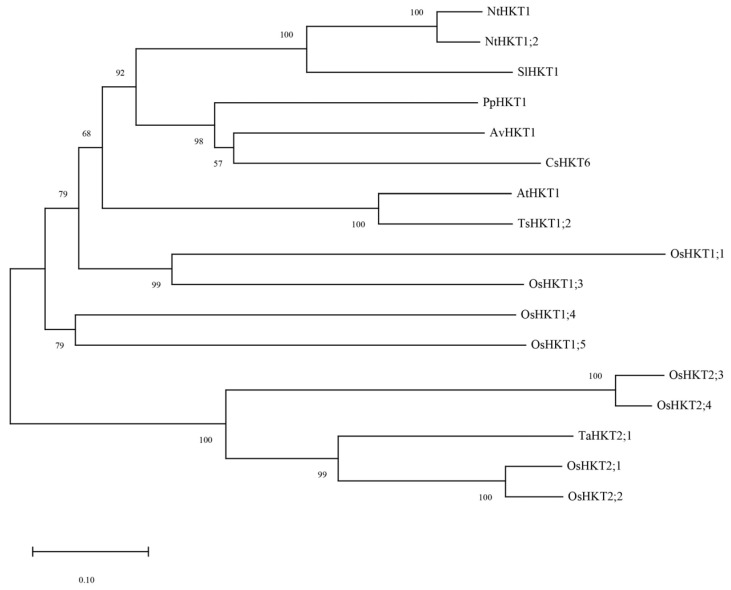
The phylogenetic analysis of AvHKT1 in kiwifruit and HKT family members from other plant species. Nt: *Nicotiana tabacum*, Sl: *Solanum lycopersicum*, Pp: *Prunus persica*, Av: *Actinidia valvata*, Cs: *Citrus sinensis*, At: *Arabidopsis thaliana*, Ts: *Thellungiella salsuginea*, Os: *Oryza sativa* subsp. *Japonica*, Ta: *Triticum aestivum*.

**Figure 2 ijms-24-15737-f002:**
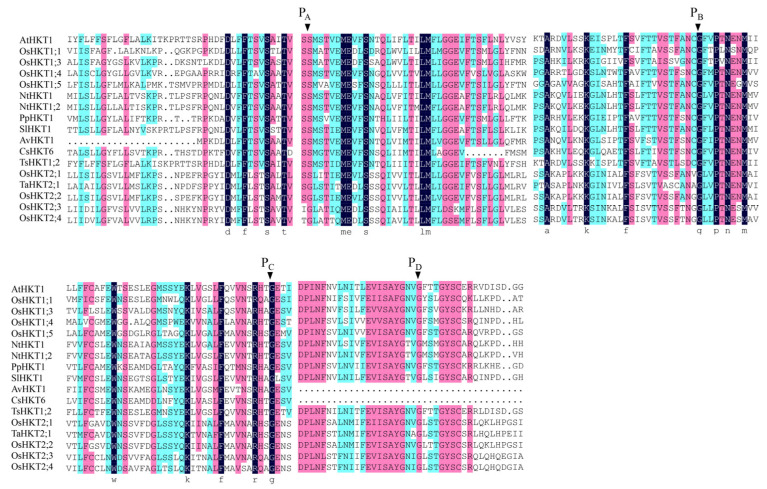
Multiple sequence alignments for HKT proteins from different plant species. Nt: *Nicotiana tabacum*, Sl: *Solanum lycopersicum*, Pp: *Prunus persica,* Av: *Actinidia valvata*, Cs: *Citrus sinensis*, At: *Arabidopsis thaliana*, Ts: *Thellungiella salsuginea*, Os: *Oryza sativa* subsp. *Japonica*, Ta: *Triticum aestivum*. The conserved Ser/Gly residues in the P_A–D_ region are indicated by the arrowhead. The identical residues are marked by different letters.

**Figure 3 ijms-24-15737-f003:**
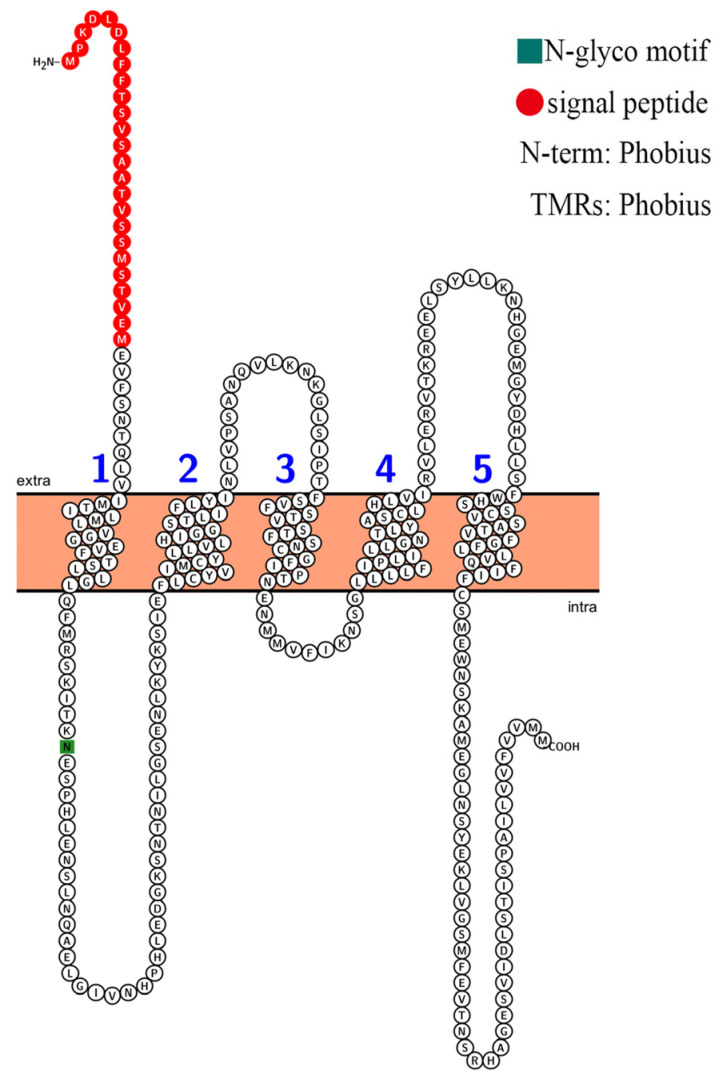
The prediction of the AvHKT1 trans-membrane domain. The different numbers represent the five transmembrane topologies.

**Figure 4 ijms-24-15737-f004:**
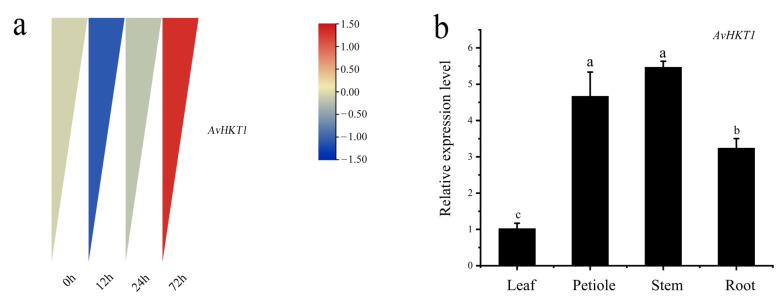
The expression pattern of *AvHTK1* in *A. valvata* under salt stress. (**a**) Analysis of *AvHKT1* expression pattern at different time points of salt stress treatment and (**b**) tissue-specific expression of *AvHKT1* under salt stress for 72 h. Different letters (a–c) represent significant mean differences at *p* < 0.05.

**Figure 5 ijms-24-15737-f005:**
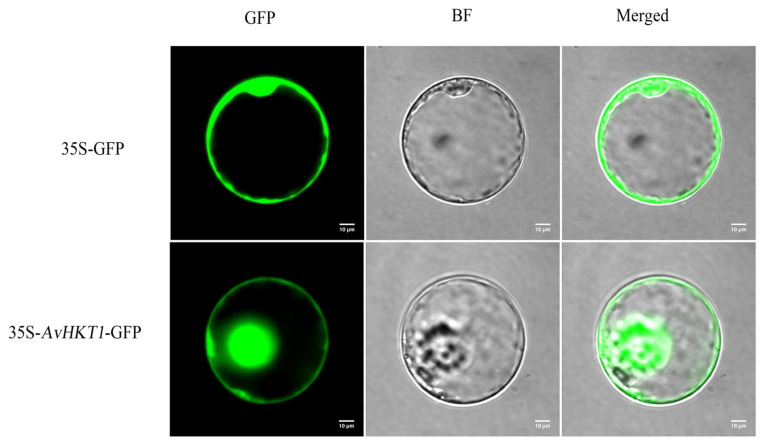
Subcellular localization analysis of the AvHKT1 protein. GFP: Green fluorescence protein; BF: Bright field; Merged: Merged field for GFP and BF. The scale bar was set at 10 µm.

**Figure 6 ijms-24-15737-f006:**
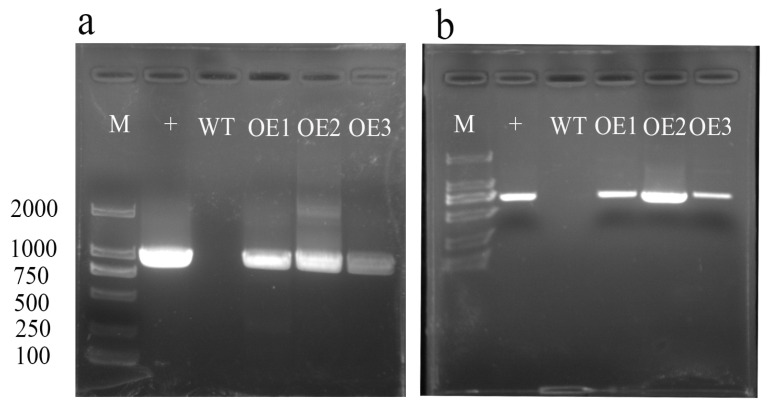
RT-PCR identification analysis of *AvHKT1* transgenic lines. (**a**) Identification of full-length *AvHKT1* sequence in overexpression lines and (**b**) detection of GFP locus fragments in overexpression vector sequences. M, 2000 bp marker; +, the recombinant plasmid containing *AvHKT1* (positive control); WT, wild-type kiwifruit; OE1, OE2, OE3, overexpression transgenic lines.

**Figure 7 ijms-24-15737-f007:**
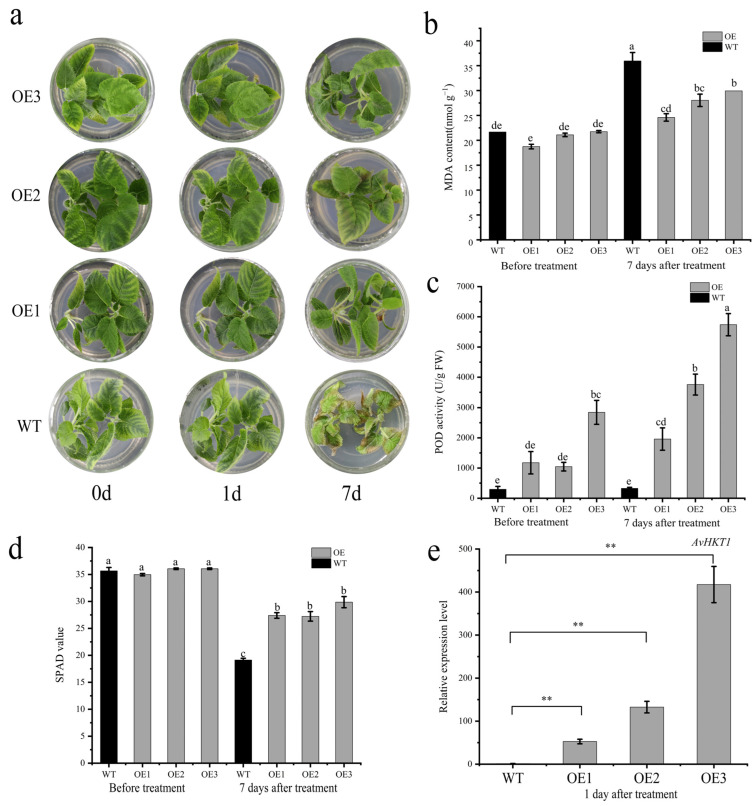
Functional validation of *AvHKT1* in kiwifruit under salt stress. (**a**) Phenotypes of WT (*A. chinensis* cv. ‘Hongyang’) and overexpression kiwifruit plants under salt stress conditions. (**b**) MDA content, (**c**) POD activity, and (**d**) SPAD value of wild-type and overexpressed plants before and after salt stress treatment. (**e**) Expression of *AvHKT1* in transgenic lines and WT plants after salt stress for 1 d. **, *p* < 0.01, (student’s *t*-test). Different letters in the same picture indicate significant differences at *p* < 0.05 level.

**Figure 8 ijms-24-15737-f008:**
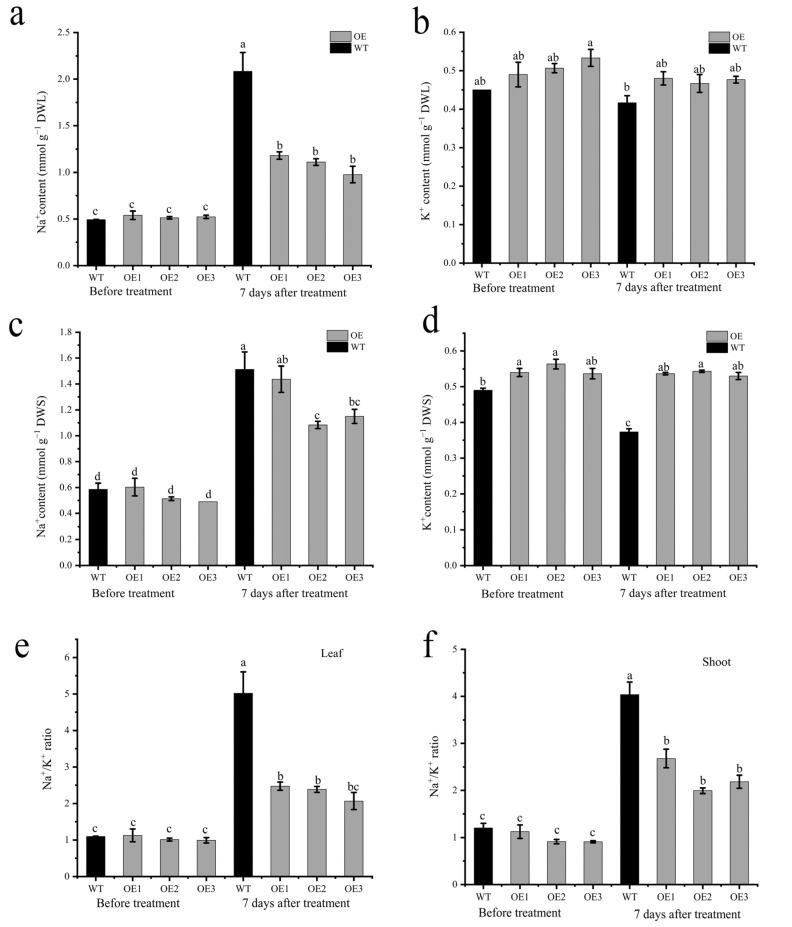
The effect of salt stress on ion content in WT and overexpressed lines. The content of Na^+^ and K^+^ in leaf (**a**,**b**) and shoot (**c**,**d**), and Na^+^/K^+^ ration in leaf (**e**) and shoot (**f**) before and after salt stress treatment. Different letters in the same picture indicate significant differences at *p* < 0.05 level.

**Table 1 ijms-24-15737-t001:** Cis-acting elements in the promoter of *AvHKT1*.

*Cis*-Acting Element	Sequence	Function Annotation
ABRE	CACGTG	Abscisic acid responsiveness
ARE	AAACCA	Anaerobic induction
Box 4	ATTAAT	Light responsiveness
CGTCA-motif	CGTCA	MeJA-responsiveness
G-Box	CACGTG	Light responsiveness
MYB	CAACAG	MYB binding site
P-box	CCTTTTG	Gibberellin-responsive element
TGA-element	AACGAC	Auxin-responsive element
TGACG-motif	TGACG	MeJA-responsiveness

**Table 2 ijms-24-15737-t002:** The ID of HKT protein in different species.

Protein	UniProt Accession
AtHKT1	Q84TI7.1
CsHKT6	A0A067FEV5
NtHKT1	A0A1S3YZS4
NtHKT1;2	A0A1S3Z6K8
OsHKT1;1	Q7XPF8
OsHKT1;3	Q6H501
OsHKT1;4	Q7XPF7
OsHKT1;5	Q0JNB6
OsHKT2;1	Q0D9S3
OsHKT2;2	Q93XI5.1
OsHKT2;3	Q8L481
OsHKT2;4	Q8L4K5
PpHKT1	A0A251QTK2
TaHKT2;1	AAA52749
SlHKT1	A0A3Q7H8N9
TsHKT1;2	BAJ34563

## Data Availability

Transcriptome raw data is submitted in NCBI public repository under BioProject (PRJNA726156).

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
