# Peer review of "A High-K+ Affinity Transporter (HKT) from Actinidia valvata Is Involved in Salt Tolerance in Kiwifruit"

_ijms, 2023, doi:10.3390/ijms242115737_

Round 1
Reviewer 1 Report
The authors studied the HKT transporter which is crucial for salt tolerance in plants. So the topic is relevant. The positive aspect of the manuscript is that they combined the bioinformatic and molecular methods. Phylogenetic analyses revealed the most and least similar species, while the multiple sequence alignment showed the conserved amino acids. Further analysis also revealed the transmembrane domains of the protein. The expression pattern of the HKT gene and its subcellular localization were also investigated. The photos and figures are clear. The promoter analysis shed light on several regulatory elements. The overexpressed lines clearly showed the role of the HKT gene in the determination of Na/K concentration.
The experiments are well-planned, and the results and conclusions are clear.
I have only a few suggestions:
Fig.1. It would be good to write in the legend the name of the species after the abbreviations. e.g. At: Arabidopsis thaliana, Nt: …..Os: ….. etc.
Fig.2. It would be good to write in the legend the name of the species after the abbreviations.
364. line: 2.5 should change 4.5.
In Fig.3. you should use larger letters for the explanation. (N-glyco motif etc.)
Table S1. The primer sequence and restriction site should also start with a capital letter.
In the reference 30. the mutant word does not need to be italicized.
Author Response
Dear Reviewer 1,
Thank you for sparing your valuable time and reviewing our MS. We have made following changes in our MS according to your valuable comments and suggestions.
Comment 1:Fig.1. It would be good to write in the legend the name of the species after the abbreviations. e.g. At: Arabidopsis thaliana, Nt: …..Os: ….. etc.
Response: we appreciate your suggestions and we have added the name of species abbreviations in Fig.1 .
Comment 2: Fig.2. It would be good to write in the legend the name of the species after the abbreviations.
Response: Thank you for your suggestions. The legend of Fig.2 have changed to‘Figure 2. Multiple sequence alignments for HKT proteins from different plant species. PA, PB, PC, and PD represent four conserved sites. Nt: Nicotiana tabacum, Sl: Solanum lycopersicum, Pp: Prunus persica, Av: Actinidia valvata, Cs: Citrus sinensis, At: Arabidopsis thaliana, Ts: Thellungiella salsuginea Os: Oryza sativa subsp. Japonica, Ta: Triticum aestivum.’.
Comment 3: 364. line: 2.5 should change 4.5.
Response: Thanks for your comments. We have changed 2.5 to 4.5.
Comment 4: In Fig.3. you should use larger letters for the explanation. (N-glyco motif etc.)
Response: We have changed the letters size of Fig.3 Legends.
Comment 5:Table S1. The primer sequence and restriction site should also start with a capital letter.
Response: We have revised it as Primer sequence and Restriction site.
Comment 6:In the reference 30. the mutant word does not need to be italicized.
Response: we have change the italicized word.
Reviewer 2 Report
Among the several statistical methods used in the manuscript, the authors used t-tests and one-way analysis of variances to make several comparisons. These two families of statistical models are based on several assumptions, e.g., normality and heteroscedasticity. I miss that the authors briefly discuss how those key assumptions were verified. For instance, judging from the error bars presented in Figures 7 and 8 (which should represent the standard errors, i.e. the square root of the variances) one might suspect that the variance homogeneity assumption fails in some cases. For instance, in Figure 8c the represented standard errors tend to increase systematically as the means increases, suggesting that alternative distributional assumptions (other than the normality) might be more appropriate (e.g., Gamma, where the variance increases quadratically with the mean, or inverse Gaussian distributions, where the variances increase proportionally to the cube of the means). These models can be easily fit using generalised linear models (implemented in any reasonable modern statistical software). I suggest that the authors to use Bartlett tests to assess the homoscedasticity.
Author Response
Dear Reviewer 2:
We are thankfull to you for preparing detail review report for our MS. We have carefully analysed our MS data according your suggestions.
We conducted bartlett analysis on all the data involved in one-way ANOVA and T- test in Figures7 and 8, and the results were as follows (Not showing all). The results of Bartlett spherical test showed that the significance P value was NaN, which was not significant at the level, and the null hypothesis was accepted, that is, the variables were independent of each other and had no correlation. All data analysis results showed that the data were independent of each other, so our analysis of variance has a certain significance. In addition, all our analyses are performed on the eight groups of data before and after treatment. In Figure.8c, the error bar of OE1 before treatment is larger than that of OE2 after treatment, but its mean is smaller than that of OE2, so the error bar does not change with the increase of the mean value.
